# Antibiotic-Resistant *Escherichia coli* Strains Isolated from Captive Giant Pandas: A Reservoir of Antibiotic Resistance Genes and Virulence-Associated Genes

**DOI:** 10.3390/vetsci9120705

**Published:** 2022-12-18

**Authors:** Siping Fan, Shaoqi Jiang, Lijun Luo, Ziyao Zhou, Liqin Wang, Xiangming Huang, Haifeng Liu, Shaqiu Zhang, Yan Luo, Zhihua Ren, Xiaoping Ma, Suizhong Cao, Liuhong Shen, Ya Wang, Liping Gou, Yi Geng, Guangneng Peng, Yanqiu Zhu, Wei Li, Yalin Zhong, Xianpeng Shi, Ziqi Zhu, Keyun Shi, Zhijun Zhong

**Affiliations:** 1Key Laboratory of Animal Disease and Human Health of Sichuan, College of Veterinary Medicine, Sichuan Agricultural University, Chengdu 611130, China; 2Leshan Vocational and Technical College, Leshan 614000, China; 3The Chengdu Zoo, Institute of Wild Animals, Chengdu 610081, China; 4Chengdu Research Base of Giant Panda Breeding, Sichuan Key Laboratory of Conservation Biology for Endangered Wildlife, Chengdu 610081, China; 5Jiangsu Yixing People’s Hospital, Yixing 214200, China

**Keywords:** *Escherichia coli*, antibiotic resistance genes, virulence-associated genes, mobile gene elements, odds ratios, captive giant panda

## Abstract

**Simple Summary:**

*Escherichia coli* (*E. coli*) carrying antibiotic resistance genes (ARGs) and virulence-associated genes (VAGs), which are considered as emerging environmental pollutants, undoubtedly causes adverse health effects on humans and animals. With the implementation of the Wild Release Plan of captive giant pandas, antibiotic-resistant *E. coli* carrying ARGs/VAGs in giant pandas’ feces that are discharged into the environment may become pollutants for nature. To evaluate the potential threat of *E. coli* strains from captive giant pandas, we analyzed the distribution and the association of ARGs and VAGs in antibiotic-resistant *E. coli*. Our results revealed a high prevalence of ARGs and a significant positive association among ARGs and VAGs, suggesting that the continuous monitoring of the impact of the spread of antibiotic-resistant *E. coli* (carried various ARGs and VAGs) is needed in future studies.

**Abstract:**

Recent studies showed that *Escherichia coli* (*E. coli*) strains isolated from captive giant pandas have serious resistance to antibiotics and carry various antibiotic resistance genes (ARGs). ARGs or virulence-associated genes (VAGs) carried by antibiotic-resistant *E. coli* are considered as a potential health threat to giant pandas, humans, other animals and the environment. In this study, we screened ARGs and VAGs in 84 antibiotic-resistant *E. coli* strains isolated from clinically healthy captive giant pandas, identified the association between ARGs and VAGs and analyzed the phylogenetic clustering of *E. coli* isolates. Our results showed that the most prevalent ARG in *E. coli* strains isolated from giant pandas is *bla*_TEM_ (100.00%, 84/84), while the most prevalent VAG is *fimC* (91.67%, 77/84). There was a significant positive association among 30 pairs of ARGs, of which the strongest was observed for *sul1*/*tetC* (OR, 133.33). A significant positive association was demonstrated among 14 pairs of VAGs, and the strongest was observed for *fyuA*/*iroN* (OR, 294.40). A positive association was also observed among 45 pairs of ARGs and VAGs, of which the strongest was *sul1*/*eaeA* (OR, 23.06). The association of ARGs and mobile gene elements (MGEs) was further analyzed, and the strongest was found for *flor* and *intI1* (OR, 79.86). The result of phylogenetic clustering showed that the most prevalent group was group B2 (67.86%, 57/84), followed by group A (16.67%, 14/84), group D (9.52%, 8/84) and group B1 (5.95%, 5/84). This study implied that antibiotic-resistant *E. coli* isolated from captive giant pandas is a reservoir of ARGs and VAGs, and significant associations exist among ARGs, VAGs and MGEs. Monitoring ARGs, VAGs and MGEs carried by *E. coli* from giant pandas is beneficial for controlling the development of antimicrobial resistance.

## 1. Introduction

*Escherichia coli* (*E. coli*) is one of the most important intestinal floras of humans and animals [1,2]. The carrying of antibiotic resistance genes (ARGs) and virulence-associated genes (VAGs) among *E. coli* undoubtedly causes adverse health effects on humans and animals [3,4]. Antibiotic-resistant *E. coli* is also considered as an emerging potential environmental pollutant [5,6,7]. ARGs and VAGs carried by *E. coli* are widely studied due to the risk of transmission to the environment [8,9,10,11]. The giant panda (*Ailuropoda melanoleuca*) is one of the endangered animals in the world; the number of giant pandas (especially captive giant pandas) has increased with the implementation of protective projects [12]. However, giant pandas are still threatened by bacterial diseases, especially the diseases caused by intestinal pathogens (such as *Escherichia coli*, *Klebsiella* spp., *Campylobacter jejuni*, *Pseudomonas aeruginosa*, *Yersinia enterocolitica* and *Clostridium welchii*) [13]. Overusing antibiotics for captive animals promotes the resistance of microorganisms [14,15,16]. *E. coli* is one of the most common microorganisms and frequently occurs in nature and the gastrointestinal tract of animals. *E. coli* is usually considered as an indicator organism for antimicrobial resistance (AMR) [7,17]. Research indicated that *E. coli* strains from giant pandas are highly resistant to antibiotics such as tetracyclines, sulfonamides and β-lactams [12,18,19]. Wang et al. showed that streptomycin was the most prevalent resistance phenotype and *strB* was the most prevalent ARG in *E. coli* strains from wild giant pandas, depicting an association between phenotype resistance and ARGs [19]. Our previous study proved a significant association between AMR and mobile genetic elements (MGEs) in the *E. coli* of captive giant pandas, while the associations of ARGs with AMR and MGEs are still unknown [12].

Previous researchers showed that an association existed between ARGs and VAGs in *E. coli* isolated from animals (yaks, weaned piglets and waterfowls) [20,21,22]. The uropathogenic *E. coli* (UPEC) isolated from patients with urinary tract infections (UTIs) showed antibiotic resistance related to particular VAGs [23]. However, the association between ARGs and VAGs in *E. coli* isolated from giant pandas is unknown. 

With the implementation of the Wild Release Plan of captive giant pandas [24,25], AMR *E. coli* carrying ARGs/VAGs in giant pandas’ feces that are discharged into the environment may increase the risk of infections in animals and humans. To better understanding the characteristics of antibiotic-resistant *E. coli*, we detected ARGs and VAGs in 84 strains isolated from clinically heathy captive giant pandas and analyzed the associations of ARGs with AMR, VAGs and MGEs in this study.

## 2. Materials and Methods

### 2.1. Bacterial Strains

Eighty-four *E. coli* strains were isolated and identified from the fecal samples of 84 clinically healthy captive giant pandas and preserved in the Key Laboratory of Animal Disease and Human Health of Sichuan (Chengdu, China) [12].

### 2.2. Antibiotic Sensitivity and Resistance 

Antibiotics for the antimicrobial susceptibility test were selected based on the information provided by breeders at Chengdu Giant Panda Breeding & Research Base (CGPBRB) and other reports about antimicrobial susceptibility in wildlife. Antibiotic susceptibility results (22 antibiotics in six categories) of all the *E. coli* strains have been reported in our previous publication [12]. The antibiotic susceptibility results were interpreted according to the CLSI criteria (Clinical and Laboratory Standards Institute, 2022).

### 2.3. DNA Extraction and Screening for ARGs and VAGs

Total genomic DNA was extracted from isolates using the TIANamp Bacteria DNA kit (Tiangen Biotech, Beijing, China), according to the manufacturer’s instructions. DNA samples were stored at −20 °C for subsequent PCR detection. We further amplified 16sRNA (16 S rRNA Primer: 5′-GAGTTTGATCCTGGCTCAG-3′; 5′-AGAAAGGAGGTGATCCAGCC-3′) [26] to confirm the quality of the extracted DNA before screening for related genes.

A total of 22 ARGs (six categories), 3 phylogenetic grouping associated genes (*chuA*, *yjaA* and *TspE4.C2*) and 17 VAGs (five categories) were screened by PCR. Phylogenetic groups (A, B1, B2 and D) were assigned as previously described by Clermont et al. [27]. Primers were synthesized by Huada Gene Technology Co., Ltd. (Shenzhen, China). The primer sequences and the amplification conditions are shown in Appendix A, as previously described [10,27,28,29,30,31]. PCR products were separated by gel electrophoresis in a 1.0% agarose gel in 1 × TAE buffer (40 mM Tris-acetate, 1 mM EDTA, pH 8.3) at 120 volts for 50 min, stained with GoldView^TM^ (Sangon Biotech, Shanghai, China) and photographed under ultraviolet light using a Bio-Rad ChemiDoc MP omnipotent imager (Bole, USA). All positive PCR products were sequenced with Sanger sequencing in both directions by Sangon Biotech (Shanghai, China). The sequences were then analyzed online using BLAST (http://blast.ncbi.nlm.nih.gov, accessed on 24 September 2022). 

### 2.4. Association Analysis between ARGs and AMR/VAGs/MGEs

A χ2 test was used to compare the association between ARGs and AMR/VAGs/MGEs. Eleven MGEs have been identified, as reported in our previous study [12]. *p*-values < 0.05 were considered to be statistically significant. Odds ratios (OR) and 95% confidence intervals were also determined for statistically significant associations. An OR < 1 and an OR > 1 represent negative and positive associations, respectively. Data analysis was performed with SPSS Statistics (version 26.0).

## 3. Results

### 3.1. Distribution of ARGs, VAGs and Phylogenetic Grouping

The patterns of the ARGs and VAGs carried by the 84 antibiotic-resistant *E. coli* strains isolated from captive giant pandas are shown in Appendix A. A total of 14 ARGs were detected from 22 ARGs in six categories (amide alcohols, aminoglycosides, β-lactams, sulfonamide, tetracyclines and quinolones). As shown in Figure 1A, the detection rate of the β-lactam resistance gene *bla*_TEM_ (100.00%, 84/84) was the highest, followed by the tetracycline resistance gene *tetA* (86.90%, 73/84) and the sulfonamide resistance gene *sul3* (61.90%, 52/84). The detection rates for the other ARGs were lower than 50%: the amide alcohol resistance genes *cmlA* (48.81%, 41/84) and *flor* (47.62%, 40/84), the β-lactam resistance genes *bla*_CTX-M_ (42.86%, 36/84) and *bla*_SHV_ (10.71%, 9/84), the sulfonamide resistance genes *sul1* (39.29%, 33/84) and *sul2* (9.52%, 8/84), the quinolone resistance genes *qnrS* (36.90%, 31/84) and *oqxAB* (13.10%, 11/84), the tetracycline resistance genes *tetC* (29.76%, 25/84) and *tetG* (2.38%, 2/84) and the aminoglycosides resistance gene *aacC2* (1.19%, 1/84). The remaining eight ARGs (*aacC4*, *aadA1*, *aphA3*, *tetB*, *tetD*, *tetE*, *cat1* and *qnrA*) were not detected in any strains. A total of 9 VAGs out of 17 in four categories (adhesion-related genes, antiserum survival factor, invasion- and toxin-related genes and iron transport-related genes) were detected in 84 strains. As shown in Table 1 and Figure 1B, the highest detection rate was *fimC* (91.67%, 77/84), followed by *ompT* (63.10%, 53/84) and *sitA* (55.95%, 47/84). The detection rates of the other VAGs were *iroN* (44.05%, 37/84), *fyuA* (39.29%, 33/84), *irp2* (38.10%, 32/84), *eaeA* (21.43%, 18/84), *cvaC* (4.76%, 4/84) and *papA* (2.38%, 2/84). The remaining eight VAGs (*tsh*, *hlyA*, *hlyF*, *iss*, *astA*, *estA*, *estB* and *elt*) were not detected in any strains. We further analyzed the distribution of VAGs related to diarrheagenic *Escherichia coli* (DEC) and extraintestinal pathogenic *Escherichia coli* (ExPEC). For the four VAGs (*eaeA*, *elt*, *estA* and *estB*) related to DEC, only *eaeA* (21.43%, 18/84) was detected in 18 strains. The detection rates of VAGs related to ExPEC (*papA*, *fimC*, *fyuA*, *iroN*, *irp2*, *sitA*, *cvaC* and *ompT*) ranged from 2.38% (*papA*, 2/84) to 91.67% (*fimC*, 77/84).

The phylogenetic grouping results showed that group B2 (67.86%, 57/84) was the most prevalent, followed by group A (16.67%, 14/84), group D (9.52%, 8/84) and group B1 (5.95%, 5/84) (Appendix A). We further analyzed the number of VAGs in different phylogenetic groups. The average number of VAGs per isolate was 0.86 in group A and 2 in groups B1 and D, and the highest average number of VAGs per isolate was 4.65 in group B2. 

### 3.2. Associations between ARGs and AMR

The distribution of resistance phenotypes and ARGs is shown in Table 2. For tetracyclines antibiotics, the resistance rates of tetracycline and doxycycline ranged from 48.81% (41/84) to 61.90% (52/84), and the detection rates of the tetracyclines resistance gene *tet* (including *tetA*, *tetC* and *tetG*) ranged from 2.38% (2/84) to 86.90% (73/84). For amide alcohols antibiotics, the detection rate of *flor* was 47.62% (40/84), and that of *cmlA* was 48.81% (41/84). The detection rate of amide alcohols antibiotics resistance genes was 21.43% for both CHL and FFC. Only one aminoglycoside-resistant gene, *aacC2* (1.19%, 1/84), was detected in this study, and the proportions of strains with aminoglycoside-resistant phenotypes ranged from 3.57% (3/84) to 10.71% (9/84). We detected a high prevalence of the β-lactam antibiotic gene *bla*_TEM_ (100%, 84/84), but only three β-lactams (AML, AMP, AMC) were found to have resistance rates higher than 46.43% (39/84). Only two of the four quinolone resistance genes were detected, and all the detection rates were below 36.90% (31/84). The resistance rates of the quinolones antibiotic ranged from 7.14% (6/84) to 10.71% (9/84). The detection rates of the three sulfonamide resistance genes ranged from 9.52% (8/84) to 61.90% (52/84), while the resistance rate of SXT was only 17.86% (15/84). The associations between ARGs and AMR were also analyzed (Appendix A). A total of 88 pairs of significant associations (*p* < 0.05) were observed between ARGs and AMR, of which 86 pairs were positively associated. The strongest was observed for CEZ and *qnrS* (OR, 24.86; 95% CI, 2.98–205.71), followed by AMP and *cmlA* (OR, 22.43; 95% CI, 4.80–104.83). Negative associations were only observed between AMC and *bla*_CTX-M_/*sul2*.

### 3.3. Associations among 13 ARGs and among 9 VAGs

Table 3 summarizes the associations among 13 ARGs (except for *bla*_TEM_ due to no results being available) detected in 84 *E. coli* strains from captive giant pandas. The results showed that a total of 30 pairs of ARGs were positively associated (OR > 1). Only one negative association was observed between *sul3* and *oqxAB* (OR, 0.18; 95% CI, 0.05–0.76). The strongest positive association was detected between *sul1* and *tetC* (OR, 133.33; 95% CI, 15.96–1113.64). Statistically significant associations (*p* < 0.05) were observed among nine VAGs (Table 4). A total of 14 pairs’ positive associations were detected among VAGs, while no negative associations were detected. The positive association between *fyuA* and *iroN* was observed as the strongest (OR, 294.40; 95% CI, 32.82–2640.87). No significant association was observed between *fimC*/*papA*/*cvaC* and other VAGs.

### 3.4. Association of ARGs with VAGs or MGEs

The association between ARGs and VAGs detected in *E. coli* strains isolated from captive giant pandas is shown in Table 5. The results showed that 45 pairs of ARGs were positively associated with VAGs. Only one negative association was observed for *tetC* and *sitA* (OR, 0.25; 95% CI, 0.09–0.64). The strongest positive association between ARGs and VAGs was observed in *sul1* and *eaeA* (OR, 23.06; 95% CI, 4.80–110.85). There was no significant association between *bla*_SHV_/*tetG*/*oqxAB*/*aacC2* and VAGs. Associations between ARGs and MGEs were further analyzed (Appendix A). A total of 46 pairs showed significant positive associations (*p* < 0.05). The strongest association was observed between *flor* and intI1 (OR, 79.86; 95% CI, 9.90–643.30), followed by *sul2* and IS26 (OR, 75.00; 95% CI, 6.73–835.97) and *cmlA* and intI1 (OR, 72.80; 95% CI, 9.07–584.18).

## 4. Discussion

Recently, antibiotic resistance *E. coli* as well as antibiotic resistance genes (ARGs) and virulence-associated genes (VAGs) carried by *E. coli* strains have been extensively studied as environmental pollutants, which may cause adverse effects to ecology, human and animal health [21,32,33,34]. To evaluate the potential threat of *E. coli* strains from captive giant pandas, the distribution and association of ARGs and VAGs in antibiotic-resistant *E. coli* were analyzed. Our results showed that many ARGs (especially β-lactam resistance genes, 100.00% for *bla*_TEM_) and VAGs (especially adhesion-related virulence genes, 91.67% for *fimC*) were detected from the 84 *E. coli* strains. Fourteen ARGs were detected, and the top five ARGs ranked by detection rates were *bla*_TEM_ (100.00%), *tetA* (86.90%), *sul3* (61.90%), *cmlA* (48.81%) and *flor* (47.62%). Another study from the Dujiangyan Giant Panda Conservation and Research Center and the Wolong Nature Reserve showed that the top five ARGs detected were *sul1* (45%), *bla*_CTX-M_ (44%), *ant(3′)-Ia* (38%), *tetA* (37%) and *qnrB* (35%) [35]. Those data showed a high prevalence of ARGs (especially for β-lactam, tetracyclines and sulfonamides resistance genes) in *E. coli* strains from giant pandas, and the carrying rate of the ARGs in the *E. coli* isolated in our study is higher than that in strains from Dujiangyan and Wolong. Notably, all of the 84 strains were positive for *bla*_TEM_ in our study. *E. coli* strains containing *bla*_TEM_ were reported in other wild animals (such as owls, foxes, wild rabbits, genets, etc.) [36]. Another study showed that 5 out of 13 ARGs were detected in *E. coli* isolated from wild giant pandas, but *bla*_TEM_ was not detected [13]. The different detection rates of ARGs (especially for *bla*_TEM_) in captive and wild giant pandas may result from their different living environments [37]. The results from non-human primates have also shown that the carrying rates of ARGs in *E. coli* are affected by living environments [10,38]. As far as we know, this is the first report of a high prevalence of *bla*_TEM_ detected in *E. coli* strains from captive giant pandas. Some studies showed that *bla*_CTX-M_ and *sul1* are prevalent in *E. coli* from captive giant pandas and wild birds [35,39]. The most prevalent ARGs in *E. coli* isolated from captive giant pandas in Dujiangyan and Wolong were *bla*_CTX-M_ (44%) and *sul1* (45%), which is consistent with our results (*bla*_CTX-M_ was 42.86% and *sul1* was 39.29%) [35]. *E. coli* carrying various ARGs was considered as an environmental pollutant, since the spread of antibiotic-resistant bacteria may pose a potential threat to human health [40,41,42]. Referring to the previous report on wild gulls, the antibiotic-resistant *E. coli* strains from giant pandas may serve as a pool of ARGs [43]. Moreover, VAGs related to diarrheagenic *Escherichia coli* (DEC) or extraintestinal pathogenic *Escherichia coli* (ExPEC) were screened [44,45,46]. In the 84 *E. coli* strains, 9 out of 17 VAGs were detected, of which the top 5 resistance genes were *fimC* (91.67%), *ompT* (63.10%), *sitA* (55.95%)*, iroN* (44.05%) and *fyuA* (39.29%). None of the isolates were positive regarding *tsh*, *hlyA*, *hlyF*, *iss*, *astA*, *estA*, *estB* and *elt*. Only *eaeA* (which is a marker of enteropathogenic *E. coli*, EPEC) was detected in 18 strains. Most strains (91.67%, 77/84) carried at least one VAG related to ExPEC, which raises a health concern. Similarly, ExPEC-associated VAGs were observed in a higher proportion than those associated with DEC in *E. coli* from waterfowl and yaks [20,21]. VAGs related to a high-pathogenicity island (*fyuA* and *irp2*) were observed in 33 strains in our study. The detection rate of the VAGs in our study (ranging from 2.38% to 91.67%) is higher than the result of Wang et.al (all detection rates below 25%) [19]. Notably, the most prevalent VAG in our study is *fimC*, which belongs to adhesion-related virulence genes. Deng et al. showed that an *E. coli* strain from a urine sample of a giant panda carried the largest number (42.45%, 239/563) of adhesion- and invasion-related virulence genes [47]. Our present study showed *E. coli* strains from clinically healthy captive giant pandas carrying a large number of VAGs, which was also observed in healthy chickens and pigs [48,49]. As the phylogenetic grouping is closely related to virulence genes (group A and group B1 carry fewer virulence genes, while group B2 and group D carry more virulence genes [31]), we further analyzed the phylogenetic grouping and the number of virulence genes carried by each *E. coli* strain from giant pandas. Our results showed that the B2 group is the most prevalent group (67.86%, 57/84) which carries the highest number of virulence genes. Another study on *E. coli* strains of wild giant pandas showed that the B2 group was the most prevalent (38%, 31/82) [13], which is consistent with our study. In summary, our results showed that the antibiotic-resistant *E. coli* derived from giant pandas is a natural reservoir of ARGs and VAGs. In our previous study, we detected a high prevalence of MGEs in *E. coli* strains isolated from giant pandas [12]. Another study also proved that the release of wild animals may spread ARGs through MGEs [50]. With the release program of captive giant pandas being implemented in China [24,25], *E. coli* strains carrying various ARGs and VAGs may be harmful when spreading to the surrounding environment, breeders and tourists, posing a higher risk to public health. 

In general, the antibiotic resistance phenotype is determined by the carrying and expression of ARGs [51,52,53]. ARGs carried by *E. coli* strains are closely related to the antibiotic resistance phenotype [54]. Thus, we further analyzed the relations between antibiotic resistance phenotypes and ARGs carried by *E. coli* from captive giant pandas. As shown in Table 2, for amide alcohols, aminoglycosides and quinolones, the antibiotic resistance phenotypes were all matched with the detection rates of ARGs. However, for tetracyclines, β-lactams and sulfonamides, the detection rates of related ARGs fluctuated (such as the resistance rate of sulfonamides being 17.86%, while the detection rate of *sul* ranged from 9.52% to 61.90%). The detection rates of ARGs are not strictly correlated with the corresponding antibiotic resistance phenotypes [10]. This might be due to the fact that the expression of ARGs is just one of the mechanisms that mediate bacterial resistance or due to the abnormal expression or low expression of ARGs that cannot cause the drug resistance [51,55,56]. Previous studies have shown an association between ARGs and VAGs carried by *E. coli* in wild waterfowls and weaned piglets [21,22]. We further analyzed the association among 13 ARGs and 9 VAGs, respectively. We observed 30 pairs of positive associations and 1 negative association among 13 ARGs, of which the strongest positive association (*p* < 0.05, OR > 1) was observed for *sul1* and *tetC*. Similarly, positive associations among different ARGs were observed in another study on wild birds [21]. For the VAGs, 14 pairs of VAGs showed positive associations, and none of the VAGs pairs were negatively associated, which is consistent with the results of *E. coli* strains from Qinghai Plateau yaks [20]. The strongest positive association (*p* < 0.05, OR > 1) was observed for *fyuA* and *iroN* within VAGs. Associations among ARGs and VAGs in *E. coli* from captive giant pandas were further analyzed. Our results showed that positive associations were more common than negative ones (45 pairs were positive and only 1 was negative). The strongest positive association was observed for *sul1* and *eaeA* between ARGs and VAGs. Interestingly, a study on waterfowls showed that negative associations were more common [21]. According to the available information, this is the first report about the association among ARGs and VAGs carried by *E. coli* from captive giant pandas. Previous studies have shown that specific microorganisms (such as endophytes or pathogenic *E. coli*) are enriched for ARGs [57,58,59], and significant associations exist between VAGs and AMR [60]. However, the direction of associations (positive or negative) among ARGs and VAGs is unknown. We speculate that carrying VAGs increases the probability of strains carrying ARGs, and vice versa. Considering that MGEs mediate the horizontal transmission of ARGs and that associations exist among MGEs and ARGs [61,62,63], we further analyzed the association between ARGs and MGEs. Only significant positive associations (*p* < 0.05, OR > 1) were observed between ARGs and MGEs. Positive associations for ARGs and MGEs were also reported in *E. coli* strains from wastewaters and activated sludge reactors [64,65]. Our present results showed that *E. coli* strains from captive giant pandas are a reservoir of ARGs, VAGs and MGEs; ARGs are significantly associated (*p* < 0.05) with VAGs and MGEs. As captive giant pandas are released into the wild, various ARGs, VAGs and MGEs carried by *E. coli* strains from captive giant pandas may have an impact on the environment.

## 5. Conclusions

The present study showed that antibiotic-resistant *E. coli* from captive giant pandas is a pool of ARGs, VAGs and MGEs, and significant associations exist among them. Positive associations are more prevalent than negative ones among ARGs and VAGs, and only positive associations were observed between ARGs and MGEs in *E. coli* from captive giant pandas. Antibiotic-resistant *E. coli* strains that carry various ARGs, VAGs and MGEs are a potential threat not only to the health of giant pandas but also to the environment. We strongly recommend implementing surveillance programs for the use of antibiotics in giant pandas and monitoring the impact of antibiotic-resistant *E. coli* (carrying various ARGs, VAGs and MGEs) on humans, other animals and the environment in future studies.

## Figures and Tables

**Figure 1 vetsci-09-00705-f001:**
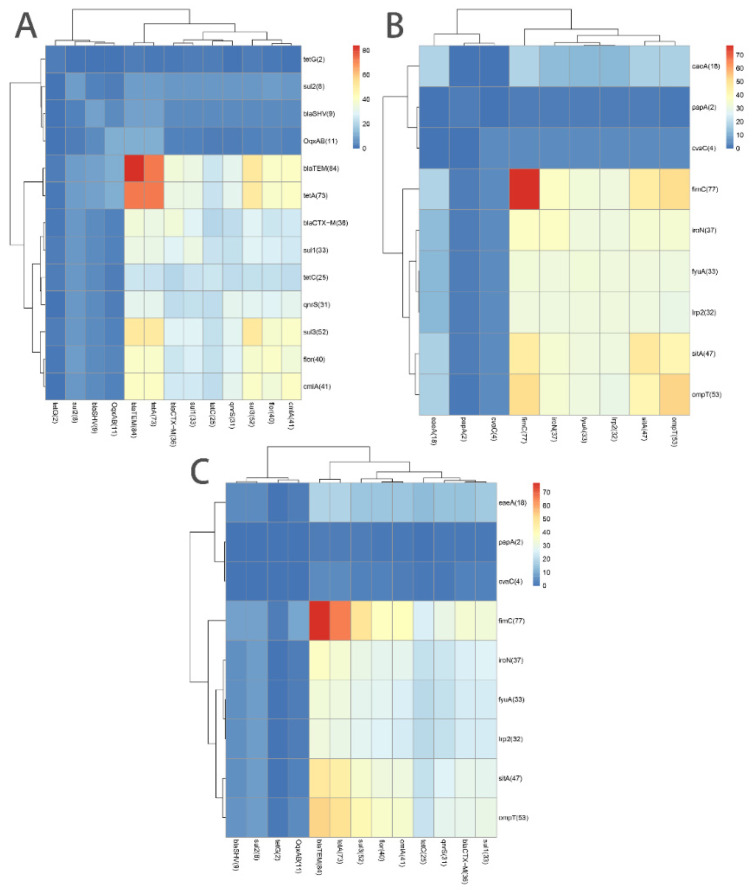
Heat map demonstrating the distribution of antibiotic resistance genes (ARGs) and virulence-associated genes (VAGs) in *Escherichia coli* (*E. coli*) isolated from captive giant pandas in Sichuan, China. The color scale and the corresponding value indicate the number of *E. coli* isolates carrying corresponding abscissa and ordinate genes. The prevalence of ARGs (**A**), VAGs (**B**), ARGs and VAGs (**C**) in *E. coli* isolates. *bla*_TEM_, *tetA* and *fimC* were the highly prevalent genes among all tested ARGs and VAGs.

**Table 1 vetsci-09-00705-t001:** Distribution of virulence-associated genes detected in *E. coli* isolated from captive giant pandas (n = 84).

Category of Virulence	Genes	No. of Positive Isolates	Percentage of Positive Isolates (%)
Adhesion-related genes	*tsh*	0	0.00
	*eaeA*	18	21.43
	*papA*	2	2.38
	*fimC*	77	91.67
Hemolysin-related genes	*hlyA*	0	0.00
	*hlyF*	0	0.00
Invasion- and toxin-related genes	*elt*	0	0.00
	*estA*	0	0.00
	*estB*	0	0.00
	*astA*	0	0.00
Iron transport-related genes	*fyuA*	33	39.29
	*iroN*	37	44.05
	*irp2*	32	38.10
	*sitA*	47	55.95
Antiserum survival factor	*cvaC*	4	4.76
	*ompT*	53	63.10
	*iss*	0	0.00

**Table 2 vetsci-09-00705-t002:** Distribution of antibiotic resistance phenotypes and antibiotic resistance genes detected in *E. coli* strains isolated from captive giant pandas (n = 84).

Category of Antibiotic	Antibiotic	Results of the Antibiotic Sensitivity Test			Antibiotic Resistance Genes	No. of Positive Isolates (%)
		No. of Resistant Ones (%)	No. of Intermediate Ones (%)	No. of Sensitive Ones (%)		
Tetracyclines	DOX	52 (61.90)	20 (23.81)	12 (14.29)	*tetA*	73 (86.90)
					*tetB*	-
					*tetC*	25 (29.76)
	TET	41 (48.81)	34 (40.48)	9 (10.71)	*tetD*	-
					*tetE*	-
					*tetG*	2 (2.38)
Amide alcohols	CHL	18 (21.43)	10 (11.90)	56 (66.67)	*flor*	40 (47.62)
	FFC	18 (21.43)	3 (3.57)	63 (75.00)	*cmlA*	41 (48.81)
Aminoglycosides	AMK	3 (3.57)	4 (4.76)	78 (92.86)	*aacC2* *aacC4* *aadA1* *aphA3*	1 (1.19) - - -
	KAN	6 (7.14)	4 (4.76)	74 (88.10)		
	STR	9 (10.71)	4 (4.76)	71 (84.52)		
	NEO	3 (3.57)	6 (7.14)	75 (89.29)		
	GM	5 (5.95)	0 (0.00)	79 (94.05)		
β-lactams	AML	39 (46.43)	35 (41.67)	10 (11.90)	*bla* _CTX-M_ *bla* _TEM_ *bla* _SHV_	36 (42.86) 84 (100.00) 9 (10.71)
	AMP	58 (69.05)	21 (25.00)	5 (5.95)		
	SAM	1(1.19)	6 (7.14)	77 (91.67)		
	AMC	68 (80.95)	7 (8.33)	9 (10.71)		
	AZM	6 (7.14)	7 (7.14)	73(86.90)		
	CRO	6 (7.14)	3 (3.57)	75 (89.29)		
	CTX	8 (9.52)	9 (10.71)	67 (79.76)		
	CEZ	7 (8.33)	4 (4.76)	73 (86.90)		
Quinolones	CIP	6 (7.14)	2 (2.38)	76 (90.48)	*qnrA*	-
	NA	9 (10.71)	1 (1.19)	74 (88.10)	*qnrS*	31 (36.90)
	NOR	7 (8.33)	0 (0.00)	77 (91.67)	*cat1*	-
	OFX	6 (7.14)	1 (1.19)	77 (91.67)	*oqxAB*	11 (13.10)
Sulfonamides	SXT	15 (17.86)	0 (0.00)	69 (82.14)	*sul1*	33 (39.29)
					*sul2*	8 (9.52)
					*sul3*	52 (61.90)

Note: DOX, Doxycycline; TET, Tetracycline; CHL, Chloramphenicol; FFC, Florfenicol; AMK, Amikacin; KAN, Kanamycin; STR, Streptomycin; NEO, Neomycin; GM, Gentamicin; AML, Amoxicillin; AMP, Ampicillin; SAM, Ampicillin/Sulbactam; AMC, Amoxicillin/Clavulanic Acid; AZM, Aztreonam; CRO, Ceftriaxone; CTX, Cefotaxime; CEZ, Cefazolin; CIP, Ciprofloxacin; NA, Nalidixic Acid; NOR, Norfloxacin; OFX, Ofloxacin; SXT, Trimethoprim-Sulfamethoxazole. - indicates a negative detection.

**Table 3 vetsci-09-00705-t003:** Odds ratios (OR) of antibiotic resistance genes in *E. coli* isolates from captive giant pandas (n = 84).

Gene (No.)	OR (95% Confidence Interval)												
	*bla*_CTX-M_ (36)	*bla*_SHV_ (9)	*tetA* (73)	*tetC* (25)	*tetG* (2)	*sul3* (52)	*sul1* (33)	*sul2* (8)	*qnrS* (31)	*flor* (40)	*oqxAB* (11)	*cmlA* (41)	*aacC2* (1)
*bla*_CTX-M_ (36)	NS	-	-	15.40 (4.55–52.12)	-	4.50 (1.66–12.25)	45.57 (12.24–169.72)	11.35 (1.33–96.99)	6.81 (2.534–18.27)	5.00 (1.96–12.74)	-	5.72 (2.21–14.81)	-
*bla*_SHV_ (9)	-	NS	-	-	-	-	-	7.00 (1.34–36.69)	-	-	14.38 (3.03–68.21)	-	-
*tetA* (73)	-	-	NS	-	-	23.18 (2.80–192.27)	7.81 (0.95–64.18)	-	NA	11.47 (1.40–94.28)	-	NA	-
*tetC* (25)	15.40 (4.55–52.12)	-	-	NS	-	11.90 (2.57–55.07)	133.33 (15.96–1113.64)	22.56 (2.60–195.77)	40.74 (10.05–165.14)	28.41 (6.03–133.98)	-	15.44 (4.11–58.03)	-
*tetG* (2)	-	-	-	-	NS	-	-	-	-	-	-	-	-
*sul3* (52)	4.50 (1.66–12.25)	-	7.81 (0.95–64.18)	11.90 (2.57–55.07)	-	NS	8.83 (2.71–28.78)	-	42.273 (5.36–333.64)	40.714 (8.58–193.17)	0.18 (0.05–0.76)	103.33 (12.74–838.13)	-
*sul1* (33)	45.57 (12.24–169.72)	-	7.81 (0.95–64.18)	133.33 (15.96–1113.64)	-	8.83 (2.71–28.78)	NS	13.46 (1.57–115.36)	12.36 (4.29–35.63)	13.15 (4.44–38.96)	-	8.91 (3.19–24.95)	-
*sul2* (8)	11.35 (1.33–96.99)	7.00 (1.34–36.69)	-	22.56 (2.60–195.77)	-	-	13.46 (1.57–115.36)	NS	15.17 (1.77–130.25)	NA	-	8.65 (1.01–73.75)	-
*qnrS* (31)	6.81 (2.534–18.27)	-	NA	40.74 (10.05–165.14)	-	42.273 (5.36–333.64)	12.36 (4.29–35.63)	15.17 (1.77–130.25)	NS	55.36 (11.41–168.57)	-	NA	-
*flor* (40)	5.00 (1.96–12.74)	-	11.47 (1.40–94.28)	28.41 (6.03–133.98)	-	40.714 (8.58–193.17)	13.15 (4.44–38.96)	NA	55.36 (11.41–168.57)	NS	-	70.20 (17.47–282.02)	-
*oqxAB* (11)	-	14.38 (3.03–68.21)	-	-	-	0.18 (0.05–0.76)	-	-	-	-	NS	-	-
*cmlA* (41)	5.72 (2.21–14.81)	-	NA	15.44 (4.11–58.03)	-	103.33 (12.74–838.13)	8.91 (3.19–24.95)	8.65 (1.01–73.75)	NA	70.20 (17.47–282.02)	-	NS	-
*aacC2* (1)	-	-	-	-	-	-	-	-	-	-	-	-	NS

Note: Only antibiotic resistance genes with a significant association (*p* < 0.05) are shown. OR, odds ratio for significant associations between antibiotic resistance genes (95% confidence interval in parentheses); NA, no results available or they could not be calculated; - indicates no significant associations (*p* ≥ 0.05); NS, no statistics were determined for the same gene.

**Table 4 vetsci-09-00705-t004:** Odds ratios (OR) of virulence-associated genes in *E. coli* isolates from captive giant pandas (n = 84).

Gene (No.)	OR (95% Confidence Interval)								
	*fimC* (77)	*papA* (2)	*fyuA* (33)	*irp2* (32)	*eaeA* (18)	*iroN* (37)	*sitA* (47)	*ompT* (53)	*cvaC* (4)
*fimC* (77)	NS	-	-	-	-	NA	-	-	-
*papA* (2)	-	NS	-	-	-	-	-	-	-
*fyuA* (33)	-	-	NS	NA	4.29 (1.42–12.98)	294.40 (32.82–2640.87)	76.80 (9.60–614.43)	20.43 (4.41–94.69)	NA
*irp2* (32)	-	-	NA	NS	4.60 (1.51–13.98)	237.67 (27.26–2072.08)	69.75 (8.74–556.40)	18.91 (4.09–87.55)	NA
*eaeA* (18)	-	-	4.29 (1.4215–12.98)	4.60 (1.51–13.98)	NS	4.55 (1.45–14.33)	20.40 (2.56–162.34)	14.17 (1.78–112.74)	-
*iroN* (37)	NA	-	294.40 (32.82–2640.87)	237.67 (27.26–2072.08)	4.55 (1.45–14.33)	NS	29.64 (7.74–113.47)	16.70 (4.48–62.30)	-
*sitA* (47)	-	-	76.80 (9.60–614.43)	69.75 (8.74–556.40)	20.40 (2.56–162.34)	29.6 (7.743–113.47)	NS	19.86 (6.19–63.64)	-
*ompT* (53)	-	-	20.43 (4.41–94.69)	18.91 (4.09–87.55)	14.17 (1.78–112.74)	16.70 (4.48–62.30)	19.86 (6.19–63.64)	NS	-
*cvaC* (4)	-	-	NA	NA	-	-	-	-	NS

Note: Only virulence-associated genes with a significant association (*p* < 0.05) are shown. OR, odds ratio for significant associations between virulence-associated genes (95% confidence interval in parentheses); the meaning of NA/NS/- is the same as that in Table 3.

**Table 5 vetsci-09-00705-t005:** Odds ratios (OR) of antibiotic resistance genes associated with virulence-associated genes in *E. coli* isolates from captive giant pandas (n = 84).

Gene (No.)	OR (95% Confidence Interval)								
	*bla*_CTX-M_ (36)	*tetA* (73)	*tetC* (25)	*sul3* (52)	*sul1* (33)	*sul2* (8)	*qnrS* (31)	*flor* (40)	*cmlA* (41)
*fyuA* (33)	8.67 (3.18–23.62)	-	14.15 (4.45–45.03)	6.30 (2.10–18.91)	12.44 (4.35–35.61)	13.46 (1.57–115.36)	7.18 (2.67–19.32)	13.15 (4.44–38.96)	6.84 (2.54–18.43)
*irp2* (32)	10.00 (3.58–27.95)	-	15.57 (4.88–50.33)	5.83 (1.94–17.49)	14.33 (4.89–42.01)	14.28 (1.67–122.50)	8.02 (2.94–21.88)	11.76 (4.00–34.59)	6.18 (2.30–16.59)
*eaeA* (18)	7.00 (2.06–23.79)	-	11.70 (3.50–39.09)	3.92 (1.04–14.84)	23.06 (4.80–110.85)	-	10.09 (2.92–34.88)	8.20 (2.16–31.18)	-
*iroN* (37)	6.82 (2.60–17.89)	-	14.11 (4.19–47.48)	4.87 (1.79–13.27)	13.51 (4.64–39.20)	10.73 (1.26–91.70)	8.01 (2.92–21.99)	9.07 (3.35–24.59)	8.14 (3.04–21.81)
*sitA* (47)	6.91 (2.51–18.98)	7.23 (1.46–35.93)	0.25 (0.09–0.64)	6.66 (2.04–21.82)	3.43 (1.37–8.62)	-	7.92 (2.63–23.90)	7.73 (2.86–20.90)	5.23 (2.04–13.44)
*cvaC* (4)	4.14 (1.52–11.27)	5.80 (1.41–23.88)	6.62 (1.79–24.55)	6.21 (2.34–16.51)	12.17 (3.29–45.06)	-	10.45 (2.83–38.63)	14.29 (4.31–47.37)	8.10 (2.82–23.31)

Note: Only antibiotic resistance genes vs. virulence-associated genes with a significant association (*p* < 0.05) are shown. OR, odds ratio for significant associations between antibiotic resistance genes and virulence-associated genes (95% confidence interval in parentheses); - indicates no significant associations (*p* ≥ 0.05).

## Data Availability

The data presented in this study are available in the article/Appendix A; further inquiries can be directed to the corresponding authors.

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
