# Peer review of "Antibiotic-Resistant Escherichia coli Strains Isolated from Captive Giant Pandas: A Reservoir of Antibiotic Resistance Genes and Virulence-Associated Genes"

_vetsci, 2022, doi:10.3390/vetsci9120705_

Round 1

Reviewer 1 Report

The article entitled ‘Antibiotic-resistant Escherichia coli strains isolated from captive giant pandas: a reservoir of antibiotic resistance genes and virulence-associated genes’ presents the ability to monitor the antibiotic-resistant strain in giant pandas.

The article presents an important and interesting issue of antibiotic resistance in animals. It is also a new way of presenting results. However, small changes should be made to improve the quality of the presented results.

Line 73- After clarification, please use the abbreviated name of the strain throughout the article.

Lines 94-96 - Too many of the same repetitions were used. It is difficult to read and confuses the reception. This snippet requires editing.

123- What were the conditions of the trial? Duration? Tension? How were the products cleaned?

 129: AMR- there is no explanation of the abbreviation

General thoughts:

Were PCR positive controls performed? If not, how was the length of the products compared? If so, how were they made?

Table 3. The unit% is given at the top, so there is no need to duplicate it still in the table.

Conclusion section: in this part, the results (conclusions of the analysis) should be described in a few sentences, please complete

Reviewer 2 Report

Siping Fan and colleagues detected antibiotic-resistant Escherichia coli strains from captive giant pandas. They found captive giant pandas had a high prevalence of ARGs and a significant positive association between ARGs and VAGs. It was very interesting to read the complete manuscript. After reading and thoroughly analyzing the data (text, figures, and table) presented by the authors, I have decided to accept this manuscript after minor revision. I have attempted to mention some of the important points to improve the quality of the paper for future submission. Please have a look at the comments/suggestions for revision:

Question 1: The language has some difficult to understand and authors are requested to check the standard of the English language to make the manuscript free of any grammar/scientific or typographic errors.

Question 2: Were the isolates also tested for colistin/polymyxin? In the current scenario, it would be interesting to search for the mcr genes.

Question 3: What are the variants of the blaCTX-M genes?

Question 4: The authors mention that with the implementation of the wild release plan of captive giant pandas, antibiotic-resistant E. coli carrying ARGs/VAGs in giant pandas’ feces that are discharged into the environment may become pollutants for nature. How about some specific suggestions on the implementation of the wild release plan of captive giant pandas?

Question 5: In this study, Escherichia coli strains from clinically healthy captive giant pandas carrying a large number of VAGs, which make a little difficult to understanding this phenomenon. I suggest in the discussion section, author should consider add some information to explain this phenomenon.

Reviewer 3 Report

Reviewer Comments

1-Line 24 you have already abbreviated E. coli, Line 25 you can start using E. coli

2-Line 44: Which TEM variants are you referring? Did you type them?

3-Line 45: “A significant positive association among 30 pairs of ARGs” with what? VAGs? Sentence needs rewording

4-Line 46: “Fourteen pairs of VAGs showed a significant positive 46 association” with what? VAgs? Sentences needs rewording

5-Line 60 needs a reference end of the sentence

6-Line 76 : why did you abbreviate Streptomycin? It is only present once in your manuscript  , remove STR

7-Line 78: Reword as following “depicting an association between phenotype resistance and AGS”

8-Line 79: add “antimicrobial”  

9-Line 82: delete also

10-Line 83: change urinary to uropathogenic

11-Line 85: delete was

12-Line 85: you have already stated an association between ARGs and VAGs in animals, including water fowls. Line 85-86. You are repeating the same association between ARGs and VAGs in water fowls. Either remove waterfowls from line 83 or you need to remove lines 85-86.

13-Line 87 is redundant line, not needed

14-Line 91: to reword , English is weak. As follows. Antibiotic resistant E. coli

15 Line 94, use abbreviation as AMR E.coli,

16-Lines94-97. TO be deleted: This is an introduction, results should be in the results section

17-Table 1: header. Replace of with about

Animals age/gender data should be aligned left

Add s to Number of Samples

18-Line 109: what do mean by kinds? To delete

19-Line 111: replace referring with according

20-Line 138, add the after by

21-Line 139: replace among out of

22-Line 139-140: rephrase the sentence
